# EnvBridge: Bridging Diverse Environments with Cross-Environment Knowledge Transfer for Embodied AI

## Abstract

In recent years, Large Language Models (LLMs) have demonstrated high reasoning capabilities, drawing attention for their applications as agents in various decision-making processes. One notably promising application of LLM agents is robotic manipulation. Recent research has shown that LLMs can generate text planning or control code for robots, providing substantial flexibility and interaction capabilities. However, these methods still face challenges in terms of flexibility and applicability across different environments, limiting their ability to adapt autonomously. Current approaches typically fall into two categories: those relying on environment-specific policy training, which restricts their transferability, and those generating code actions based on fixed prompts, which leads to diminished performance when confronted with new environments. These limitations significantly constrain the generalizability of agents in robotic manipulation. To address these limitations, we propose a novel method called EnvBridge. This approach involves the retention and transfer of successful robot control codes from source environments to target environments. EnvBridge enhances the agent's adaptability and performance across diverse settings by leveraging insights from multiple environments. Notably, our approach alleviates environmental constraints, offering a more flexible and generalizable solution for robotic manipulation tasks. We validated the effectiveness of our method using robotic manipulation benchmarks: RLBench, MetaWorld, and CALVIN. Our experiments demonstrate that LLM agents can successfully leverage diverse knowledge sources to solve complex tasks. Consequently, our approach significantly enhances the adaptability and robustness of robotic manipulation agents in planning across diverse environments.

## 1 Introduction

The development of Large Language Models (LLMs) has remarkably advanced various fields, demonstrating impressive capabilities in understanding and generating human-like text. Methods such as Chain of Thought (CoT) (Wei et al. (2023)) and ReAct (Yao et al. (2023)) have showcased the high reasoning capabilities of LLMs, while models like OpenAI-o1 have further enhanced these abilities. Building on these strengths, researchers have explored the integration of LLMs into intelligent agents capable of complex decision-making tasks. These LLM agents leverage sophisticated language understanding and reasoning abilities to perform a wide range of functions, from customer service chatbots to complex problem-solving systems, demonstrating their versatility and potential.

Among the various applications of LLM Agents, robotic manipulation stands out as a particularly promising area. Recent studies such as EmbodiedGPT (Mu et al. (2023)) combine LLM with visual components and policy net, building agents based on LLM's planning ability. Methods such as Code as Policies (CaP) (Liang et al. (2023)) and VoxPoser (Huang et al. (2023)) utilize LLMs to create robotic control code, achieving high accuracy and stability in responding to instructions and removing the need for extensive pre-training specific to robot actions. However, despite these advancements, LLM embodied agents still face significant limitations. LLM agents which use natural language for planning often require policy training to align text action steps with the environment's numerical action space. This need for environment-specific policy training restricts their flexibility

to adapt to diverse and new environments. On the other hand, LLM agents that generate control code for robots, while eliminating the need for pre-training, still exhibit poor performance when applied to unseen environments due to the use of fixed, human-created prompts. This shortcoming reveals their limited transferability and lack of robustness across diverse settings. Essentially, current methods struggle to support continuous learning and adaptation, which are crucial for sustained high performance in changing environments.

Our research addresses these critical limitations by introducing a novel agent, EnvBridge. EnvBridge employs a key process called Cross-Environment Knowledge Transfer to integrate insights from different environments. This is based on traditional human intelligence: gaining insights from successful experiences in previous familiar environments and transferring knowledge to similar tasks in unseen scenarios. In a similar approach, we leverage successful robotic control code from one benchmark environment and apply it to another, thereby enhancing the agent's adaptability and performance in diverse environments. This approach alleviates the environmental constraints, providing a more flexible and generalizable solution.

EnvBridge is comprised of three components: Code Generation, Memory-Retrieval, and Re-Planning with Transferred Knowledge. When encountering the tasks in the target environment, EnvBridge will first use LLMs to generate robotic control code attempting the task. If the task is not accomplished, EnvBridge will retrieve successfully executed codes solving similar tasks in source environment stored in our memory. Subsequently, EnvBridge transfers the codes into knowledge appropriate for the target environment and applies them as in-context learning for the agent. By leveraging on appropriate transfer of knowledge across environments, EnvBridge can operate across diverse environments with high performance.

We evaluate EnvBridge across three representative embodied environments: RLBench, MetaWorld, and Calvin. These benchmarks encompass common actions like pick and place, while each presents unique environments and a diverse range of tasks. EnvBridge achieves an average success rate of 69% on RLBench tasks, significantly outperforming both code generation and Re-Planning baselines. Furthermore, EnvBridge demonstrates robust performance across various knowledge sources and task instructions, highlighting its adaptability to diverse environments and tasks.

Our contributions can be summarized as follows:

- We propose a novel agent named EnvBridge, which tackles the key challenges in LLM embodied agents that struggle with transferability using a process called Cross-Environment Knowledge Transfer. Inspired by human intelligence—where successful experiences in familiar environments are applied to similar tasks in new scenarios—EnvBridge leverages successful examples from one benchmark environment and applies them to another, enhancing adaptability and performance.

- EnvBridge enhances adaptability across various environments without relying on pre-training or human-initiated prompt adjustments. It uses Cross-Environment Knowledge Transfer to retrieve successful control codes from memory and adapt them for the target environment, allowing the agent to flexibly handle new environments and tasks.

- We evaluate EnvBridge with three distinct robot manipulation benchmarks: RLBench, MetaWorld, and CALVIN. Each benchmark has different settings and task complexities, making them ideal for evaluating EnvBridge's performance. Our results demonstrate that EnvBridge consistently achieves high task success rates across varied scenarios, confirming its effectiveness and robustness.

## 2 RELATED WORK

### 2.1 ADVANCEMENTS IN REASONING WITH LLMS

LLMs have made significant strides in natural language processing tasks, including text generation, summarization, translation, and tasks that require elements of reasoning such as reading, comprehension, and question answering. Though their capacity for reasoning and complex problem-solving is impressive, there is still much room for improvement. To enhance their reasoning skills, researchers are exploring techniques like hierarchical reasoning, where complex problems are broken down into smaller sub-problems. Wei et al. (2023) developed chain-of-thought prompting, a strategy that

prompts LLMs to reason by articulating their thought process step-by-step. Multi-agent discussion, as demonstrated in Debate (Du et al. (2023)), involves multiple LLMs working together to reason collaboratively. To further improve LLM reasoning, incorporating external knowledge is crucial. Approaches like knowledge graph integration (Pan et al. (2024)) and Retrieval-Augmented Generation (Lewis et al. (2021)) allow LLMs to access and leverage structured information, leading to more accurate and informed responses.

## 2.2 LLM AGENTS WITH MEMORY

LLM-based agents can interact with humans using natural language, offering a more flexible and transparent interface. The memory module is crucial for enabling an agent to acquire, accumulate, and use knowledge through interactions with its environment, achieved via three key operations: memory reading, writing, and reflection. Memory reading extracts useful information from past experiences to guide future actions, while memory writing stores environmental data for later use. Memory reflection allows agents to evaluate their cognitive processes. Unlike traditional LLMs, agents must learn and act in dynamic environments, necessitating a module that aids in planning future actions. RAP (Kagaya et al. (2024)) dynamically leverages past experiences corresponding to the current situation and context, thereby enhancing agents' planning capabilities. SayPlan (Ahn et al. (2022)) is an embodied agent tailored for task planning. In this system, scene graphs and environmental feedback function as the agent's short-term memory, directing its actions. The Generative Agent (Park et al. (2023)) utilizes a hybrid memory structure to support the agent's behavior. Its short-term memory holds contextual information about the agent's present situation, while its long-term memory stores past behaviors and thoughts, which can be accessed based on current events.

## 2.3 EMBODIED AGENTS

Recent works have developed more efficient reinforcement learning agents for robotics and embodied artificial intelligence to enhance agents' abilities for planning, reasoning, and collaboration in embodied environments. Hierarchical planning models (e.g., Sharma et al. (2022)) represent a type of approach where a high-level planner leverages LLMs to generate subgoals, and a low-level planner carries out their execution. The LLMs outline a sequence of subgoals aimed at reaching the final objective specified by the language instruction, while the low-level planner translates these subgoals into appropriate actions for the given environment. Dasgupta et al. (2023) proposes a similar idea of a unified system for embodied reasoning and task planning, combining high-level commands with low-level controllers for effective planning and action execution. Furthermore, various approaches that utilize the high code-generation capabilities of LLMs for robotic manipulation are being researched. These include methods for grounding actions available within the environment (Singh et al. (2022)), attempting error recovery in case of failures (Duan et al. (2024)), and learning skills from a lifelong learning perspective (Tziafas & Kasaei (2024)). These methods do not require pre-training and can take advantage of the flexible reasoning abilities of LLMs, making them highly notable. However, their applications across various environments are still limited.

# 3 ENVBRIDGE: CROSS-ENVIRONMENT KNOWLEDGE TRANSFER AND RE-PLANNING

In this section, we introduce how we design and combine Cross-Environment Knowledge Transfer, Re-Planning method with LLM code generation to build our agent EnvBridge. Figure 1 provides an overview of the framework. The framework consists of three main components: Code Generation, Memory Retrieval, Knowledge Transfer and Re-Planning.

## 3.1 PROBLEM FORMULATION

In this work, we consider an agent operating in a particular environment $E_0$ and assigned with completing some task given a free-form language instruction $l$. To achieve this goal, the agent needs to generate robot trajectories $\tau_l$ according to $l$ .

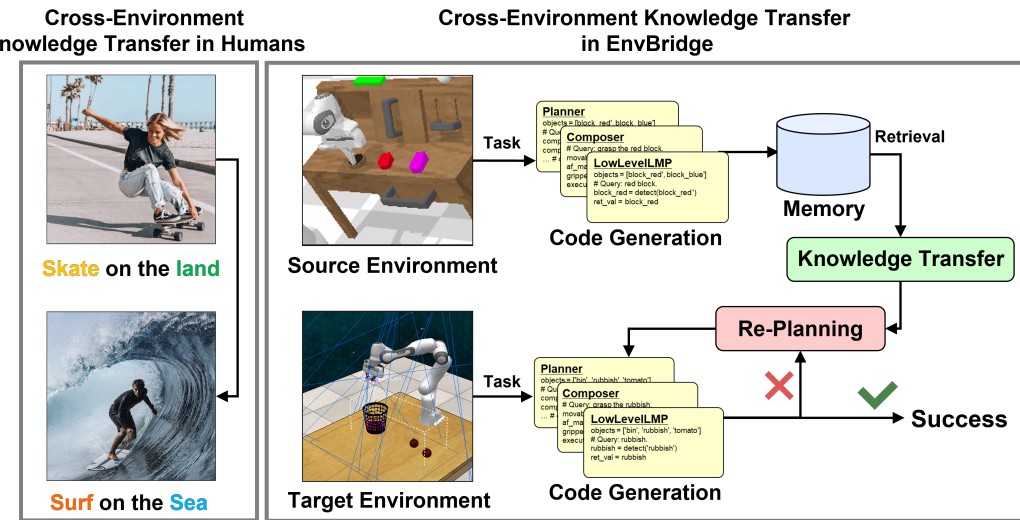

Figure 1: Overview of EnvBridge. In the method for generating code for robot operation, the successfully generated code in the source environment is stored in memory and utilized for code generation (Re-Planning) in the target environment. At that time, codes with high query similarity are retrieved from memory and converted to match the style of the target environment, thereby facilitating smooth knowledge transfer.

## 3.2 CODE GENERATION FOR ROBOT MANIPULATION

We use LLMs to generate Python code that is executed by an interpreter to decompose the language instruction into subtasks, invoke perception APIs and generate robot trajectories. Our Code Generation pipeline follows the design from VoxPoser (Huang et al. (2023)), which comprises 3 LLM-calling steps:

Step 1: The Planner is responsible for decomposing language-instruction guided tasks (e.g., "press the light switch") into sub-tasks (e.g., "grasp the button", "move to the center of the button").

$$Planner(l) = (l_1, l_2 ... , l_n) \tag{1}$$

Step 2: The Composer takes in sub-task instruction $l_i$ (e.g., "grasp the button", "move to the center of the button") and invokes necessary low-level language model program (LMP). Each low-level LMP is responsible for unique functionality (e.g., parsing query objects in dictionary form, generating affordance map in Numpy array from Composer parametrization).

$$Composer(l_i) = (LMP_1(l_i), LMP_2(l_i), .. LMP_k(l_i)) \tag{2}$$

Step 3: The Low-Level LMPs will be executed to interact with certain perception APIs to generate necessary value maps such as affordance and avoidance maps. These value maps will then be used together to find a sequence of end-effector positions serving as robot trajectories.

$$exec_{LMP}(Composer(l_i)) = \tau_{li} \tag{3}$$

Thereafter, we combine the robot trajectories of each sub-task $l_i$ to form the complete robot trajectories attempting to accomplish the task.

$$\tau_l = (\tau_{l_1}, \tau_{l_2} ... , \tau_{l_n}) \tag{4}$$

Additionally, the prompt used for code generation is documented in Appendix A.2.1.

## 3.3 MEMORY CONSTRUCTION AND RETRIEVAL

Although Code Generation method can be applied to different embodied environments, we observe a high failure rate in solving tasks in unseen environments. The challenges mainly come from two

perspectives. The high-level planner struggles to decompose unseen tasks in new environments and the composer fails to assign proper low-level LMPs. These reveal LLM's limitations when reasoning in complex zero-shot scenarios. Aiming to build a robust embodied agent that can function in various environments with high efficacy, we leverage on traditional human behavior: learning from successful experiences in previous familiar environments and transferring knowledge to similar tasks in unseen scenarios. To enhance the ability for our agent, we build our Memory and Retrieval mechanism, inspired by the Retrieval-Augmented Generation (Lewis et al. (2021)) framework.

### 3.3.1 MEMORY COMPONENT

Here, we describe in detail the contents stored in the memory component shown in Figure 1. During the execution of tasks in the Code Generation pipeline, our system records the details of each task, including the environment, the input instructions, the generated code, and the outcomes (successes or failures). After execution and evaluation, only successfully executed codes are saved into our memory. We place heavier emphasis on Planner-level and Composer-level code as they are highly related to the specific task instruction and can be flexible, while low-level LMPs code has less variation in terms of instructions. Therefore, we save only Planner and Composer-level code in our memory.

Each successful log $L$ contains the following information: The environment E, the code-level $c$ (Planner or Composer), the instruction $l$ (task instruction on Planner-level and decomposed sub-task instruction on Composer-level), and the successfully executed code. Examples of the information recorded in memory are provided in Appendix A.3, showcasing the types of data our system stores.

$$L_{planner} = \{E, c = planner, l, (l_1, l_2..., l_n))\} \tag{5}$$

$$L_{composer} = \{E, c = composer, l_i, (LMP_1(l_i), LMP_2(l_i), ..LMP_k(l_i))\} \tag{6}$$

### 3.3.2 MEMORY RETRIEVAL

When encountering a new task $T$, EnvBridge can query the memory to find relevant successful logs that can be adapted and helpful at the two stages of code generation.

Assuming we are at the code generation stage $c_T$, the instruction is $l_T$. For each log in the memory $L_j = \{E, c_j, l_j, Code_j\}$, we will calculate a similarity score between the instructions from the memory log and the current instruction of task $T$.

$$Score(L_j) = \mathbf{1}_{\{c_T = c_j\}} * cos\_sim(l_T, l_j) \tag{7}$$

Here, the similarity score is calculated as the cosine similarity between the text embedding of current task instruction and logged instruction, if they are on the same code generation level ($c_T = c_j$); otherwise, it is 0. Then we retrieve the codes with the highest similarity scores. For generating these text embeddings, we employ sentence-transformers (Reimers & Gurevych (2019)).

After retrieval, EnvBridge transfers the retrieved codes and applies them as prompts for agents to generate more accurate code in the current task. This will be further explained in Section 3.4.

$$Retrieved(Code_1, Code_2, ..Code_K) = TOP_K^{Score(L_j)}\{Code_j\} \tag{8}$$

### 3.4 RE-PLANNING WITH TRANSFERRED KNOWLEDGE

In this section, we introduce two important designs to apply retrieved code from source environments to LLM code generation in the target environment: Knowledge Transfer and Re-Planning. In Section 3.4.1, we introduce how Knowledge Transfer minimize the difference in robotic manipulation code between source and target environments. In Section 3.4.2, we introduce the instances and the form we apply the adapted retrieved code to better leverage in-context learning for code generation.

### 3.4.1 KNOWLEDGE TRANSFER

Robot foundation models such as RT-X (Collaboration (2024)) learn by integrating diverse robotic manipulation data, enabling them to operate in various environments. This clearly demonstrates that

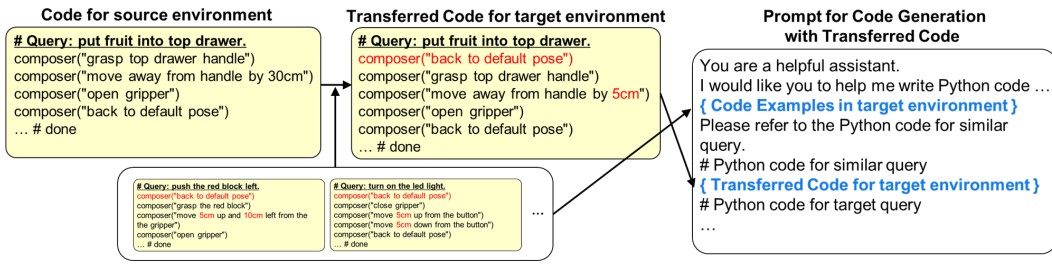

Figure 2: The process flow from Knowledge Transfer to Re-Planning. From left to middle: Using a code example from the target environment as a reference, convert the code from the source environment. From middle to right: Use the target environment's example and the transferred code to generate new code for the task.

---

**Algorithm 1** Re-Planning in EnvBridge

    **Initialize:** Environment $E$, Memory $M$, MaxTrial $d$
    **Input:** Task $T$, Instruction $l$
    $code$ = CodeGeneration($l$, $E$) // Code Generation for 1st trial
    $result$ = Execution($code$, $E$)
    **if** $result$ is Fail **then**
        **for** $i$ in range($d$ - 1) **do**
            $code_m$ = MemoryRetrieval($l$, $M$, $i$) // Memory Retrieval for $i$th similar code
            $code'_m$ = KnowledgeTransfer($code_m$) // Knowledge Transfer into target environment
            $code$ = CodeGeneration($l$, $code'_m$, $E$) // Code Generation with transferred code
            $result$ = Execution($code$, $E$)
            **if** $result$ is Success **then**
                break
            **end if**
        **end for**
    **end if**

---

common information is shared across different tasks and environments. Building on these insights, we apply Cross-Environment Knowledge Transfer for robotic planning by adapting the retrieved code to different target environments. This process minimizes differences between environments and focuses on transferring the essential insights needed for the target context.

This process is illustrated in the left and center of Figure 2. Here, code examples from the target environment are provided as prompts, and the retrieved code is adapted to suit the target environment by LLMs. This allows us to convert environment-specific plans (such as initializing a robot arm at the beginning) and environment-dependent information like coordinates and scales to match the target environment. The prompt used for knowledge transfer is provided in Appendix A.2.2.

### 3.4.2 RE-PLANNING

Using the code adapted by Knowledge Transfer, EnvBridge proceeds with Re-Planning. Figure 2 shows this process progressing from the middle to the right. If the initially generated code fails, the agent utilizes the most similar retrieved code in LLM inference context to generate new code. If this new attempt also fails, the agent uses the second most similar code to generate the code again. In this manner, EnvBridge incorporates new ideas from similar knowledge to perform Re-Planning. By combining initial prompt examples with those retrieved from memory, we leverage in-context learning. This approach allows us to maintain code quality while integrating new insights.

This methodology is akin to the technique for generating new ideas discussed by Lu et al. (2024) and demonstrates that leveraging archived ideas or insights for the target task is a viable approach. Consequently, this method enhances the agents' capabilities, improving their flexibility and creativity in task resolution. Detailed prompt illustrating this process is provided in Appendix A.2.3.

# 4 EXPERIMENTS

To assess the effectiveness of EnvBridge, we conducted experiments across three representative embodied agent benchmarks: RLBench, MetaWorld, and CALVIN.

These benchmarks are chosen for their widespread use in embodied agent research and their near-real-world robotic settings. These benchmarks share common actions, such as pick and place, but they each include different variations in terms of robots and types of tasks. In the following section, we present the evaluation results, demonstrating how memory transfer—where knowledge gained in one environment is applied to an unfamiliar one—enhances the agent's efficiency and performance. This approach allows us to assess the agent's adaptability to new challenges while leveraging previously learned knowledge to solve novel tasks, closely mimicking real-world problem-solving scenarios for embodied AI.

## 4.1 RLBENCH: MAIN RESULTS COMPARED WITH BASELINES

RLBench(James et al. (2019)) is a robot learning benchmark for tabletop environments with various tasks. We sampled 10 tasks from it and conduct evaluations covering various tasks, instructions, and objects. Each task comprises 20 trials, with instructions randomly selected from those supported by RLBench. GPT-4o-mini was used for this evalution.

We compare EnvBridge with Voxposer(Code Generation baseline) and two Re-Planning baselines: Retry and Self-Reflection. Retry indicates re-executing without making any changes if the previous trial fails. As the method for Self-Reflection, we used a novel conventional approach (Shinn et al. (2023)) to generate a plan for the next trial. Additionally, to extend Self-Reflection to RLBench, we create the prompts ourselves and use the images obtained from RLBench as Observations to generate the next plan. The prompts used for generating Self-Reflection are described in the Appendix A.2.4. In the proposed method, we conduct the evaluation using memory constructed from successful codes in the CALVIN benchmark (Mees et al. (2022)). While detailed settings for the CALVIN benchmark are described in Section 4.3, we used the successful codes from 200 tasks in CALVIN, executed with the Retry Re-Planning method, to build the memory. Moreover, for methods excluding the baseline, the success rate is calculated based on up to three trials.

The evaluation results are shown in the Figure 3. From these results, our proposed method performs better than Self-Reflection. Furthermore, while our method requires memory from other environments, Self-Reflection necessitates manual creation of prompts for reflection for each environment. Therefore, our method is more generalized for expanding the application range of robotic operations.

When analyzing individual task performance, we observe distinct trends between Self-Reflection and EnvBridge. Self-Reflection shows improved performance on tasks partially addressed by the baseline, such as BeatTheBuzz and CloseDrawer. In contrast, EnvBridge demonstrates significant improvements on tasks with low baseline performance, like PushButton and TakeLidOffSaucepan. This difference arises because Self-Reflection enhances the generation of better code for tasks that are already partially solvable, whereas EnvBridge can solve previously unsolvable tasks by incorporating new insights from other environments. Additionally, specific examples of success and failure with EnvBridge are provided in Appendix A.5.

## 4.2 METAWORLD: EVALUATION WITH VARIOUS KNOWLEDGE SOURCES

MetaWorld (Yu et al. (2021)) is an open-source simulated robotics environment designed for multi-task and meta-learning scenarios. It provides a suite of benchmark manipulation tasks of varying complexity, all sharing a common robot arm and basic physics. MetaWorld's standardized task set and simulated table robotic manipulation scenario are similar to the settings in RLBench, making it an appropriate benchmark environment to evaluate EnvBrdige's transferability across environments.

We select five tasks from MetaWorld's standardized task set for evaluation. Assembly, Button-Press, Drawer-open, Hammer, and Window-close. One instruction is given for each task. For each task, we evaluate with 20 trials. GPT-4o is used for this evalution.

For MetaWorld, we evaluate EnvBridge's performance with different sources of knowledge. They are in-domain knowledge (memory from MetaWorld), transferred knowledge (memory from RL-

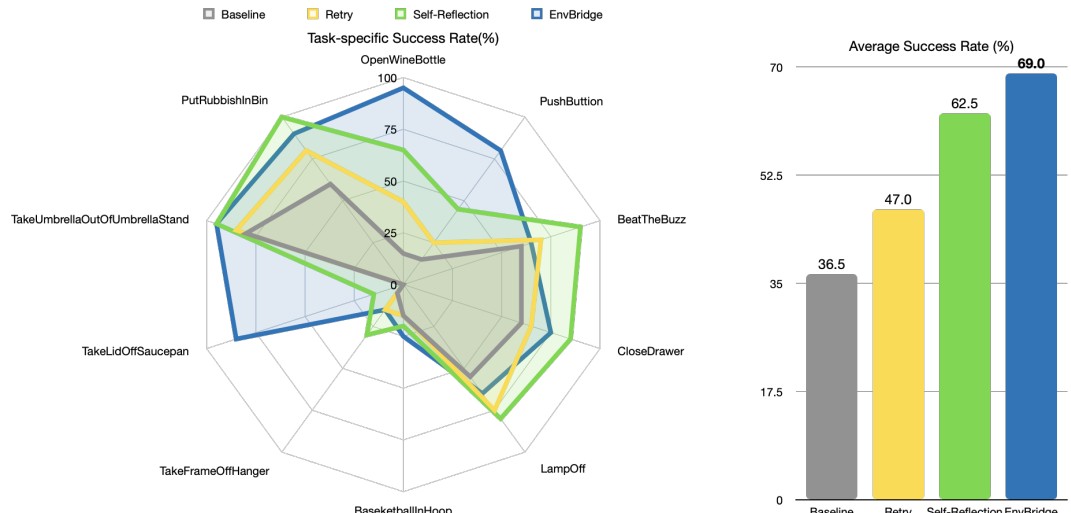

Figure 3: Task-specific and average success rate(%) on RLBench.

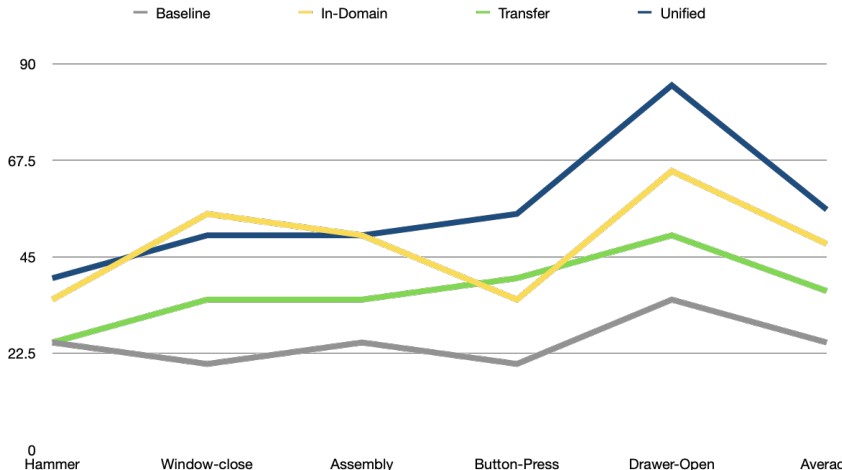

Figure 4: Task-specific success rate(%) on MetaWorld

Bench) and unified knowledge (a combined memory from MetaWorld and RLBench). We compare EnvBridge under various settings with Voxposer, which is the LLM code generation agent baseline.

From Figure 4 we can see that with transferred knowledge from RLBench, EnvBridge can surpass the perfomance of robotic manipulation code generation baseline, which proves EnvBridge's efficacy in the unseen environment. Furthermore, if EnvBridge can retrieve successful experiences from unified memory of the source environment and target environment, it can have a better performance than the agent with only in-domain knowledge, which further prove EnvBridge's robustness.

## 4.3 CALVIN: EVALUATION ON INSTRUCTION ROBUSTNESS

CALVIN (Mees et al. (2022)) is a robot manipulation benchmark for language-conditioned tasks. It allows for the construction of long-horizon tasks, which are specified solely through natural language. To maintain a consistent number of subtasks, we chose 200 single tasks from the validation dataset instead of long-horizon tasks, with GPT-4o-mini as the LLM for the evalution. For comparison, we use the learning-based model provided by CALVIN, known as multi-context imitation learning (MCIL) (Lynch & Sermanet (2021)), as well as two Re-Planning methods: Retry and EnvBridge. Additionally, CALVIN includes 15 types of tasks, primarily involving the movement of blocks, with each type having a single instruction assigned to it. To assess robustness to variations in instructions, we conducted two types of evaluations. The first evaluation uses a single instruction,

specifically the one provided by CALVIN. The second evaluation uses multiple instructions, where we generated five additional paraphrased instructions for each task using GPT-4o.

As shown in Table 1, the results indicate that in the evaluation with a single instruction, Retry method performs slightly better. However, when evaluated with paraphrased instructions, the performance of Retry degrades, whereas our method demonstrates better performance. These results indicate that while the effect is less observable in environments with limited variations in instructions and tasks, such as CALVIN with single instructions, our method is confirmed to be effective in more complex environments with greater variations, like CALVIN with paraphrased instructions and RLBench.

Table 1: Success rate(%) on CALVIN.

| Method | Single Instruction | Paraphrased Instructions |
|--------|--------------------|--------------------------|
| MCIL   | 32.0               | -                        |
| Retry  | 61.0               | 57.5                     |
| Ours   | 60.5               | 63.0                     |

## 5  ABLATION STUDY

### 5.1  KNOWLEDGE TRANSFER

In EnvBridge, Knowledge Transfer is a crucial process for connecting different environments. Therefore, we evaluated its impact on performance. In this evaluation, we assessed the scenario without Knowledge Transfer described in Section 3.4.1, directly using the code from different environments for Re-Planning. The results are shown in Figure 5. These results confirm that Knowledge Transfer improves performance and enables smooth transfer of insights across various environments.

### 5.2  MEMORY RETRIEVAL

In the Memory-Retrieval mechanism, it is important to efficiently utilize the various codes stored in memory. In this section, to verify the effectiveness of our similarity-based query retrieval, we compare it with the scenario where codes are randomly selected. The results are shown in Figure 5. These results demonstrate that our method improves performance compared to random choice, confirming that selecting the most relevant codes will effectively improve the performance.

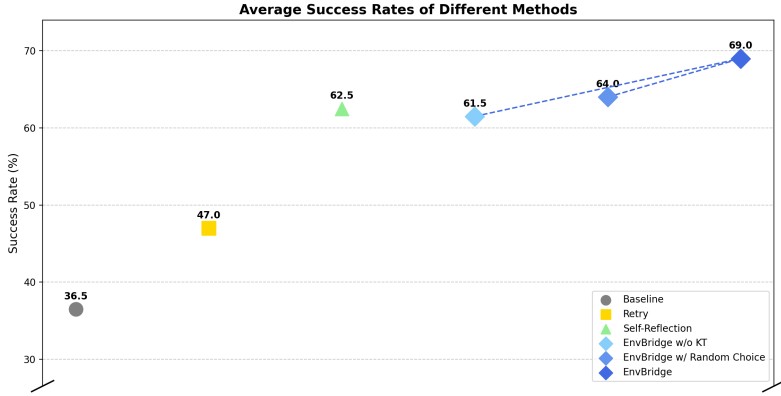

Figure 5: Performance comparison across different methods on RLBench.

### 5.3  MEMORY COMPARISON

Memory is an important component in EnvBridge, and performance varies depending on the codes stored. We evaluated the performance differences due to different memories. Specifically, we constructed and assessed memories from three different benchmarks: RLBench, MetaWorld, and

CALVIN. The results are shown in Table 2. All results with EnvBridge show improved performance compared to Retry. Interestingly, the performance with memory from CALVIN outperformed that with memory from RLBench, even though RLBench is the same environment used in the evaluation. On the other hand, the performance using memory from MetaWorld is worse compared to the other two. Upon analysis, it is found that MetaWorld's memory contains 10 codes as planners, whereas CALVIN has 26 and RLBench has 50 codes. It is believed that the smaller variation in the codes stored in the memory affects the performance. This suggests that incorporating various knowledge from other environments can potentially yield better results.

## 5.4 ACCURACY IMPROVEMENT WITH TRIALS

EnvBridge promotes the generation of new code by incorporating insights from different environments. To verify whether EnvBridge enables effective Re-Planning, we examine trends in accuracy improvement with the number of trials. The results are shown in Figure 6. The figure illustrates that the accuracy of the EnvBridge method improves over the number of trials and consistently outperforms the Retry method. This demonstrates that EnvBridge is more effective in achieving accurate results through its Re-Planning strategy.

Table 2: Performance comparison of different memory on RLBench.

| Memory | Success rate(%) |
|---|---|
| Retry without memory | 47.0 |
| MetaWorld | 57.5 |
| CALVIN | 69.0 |
| RLBench | 65.5 |

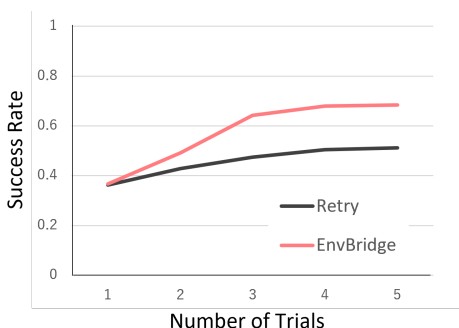

Figure 6: Accuracy improvement with number of trials on RLBench.

## 6 CONCLUSION AND LIMITATIONS

We propose EnvBridge, a novel method for transferring knowledge from various environments to target environment in robotic manipulation agents. EnvBridge stores successful control codes from one environment in memory and transfers this knowledge to other environments to apply it, thereby enhancing the agent's adaptability and performance. Our approach demonstrates superior performance in robotic manipulation benchmarks.

While EnvBridge shows promising results, several important challenges remain. Our current implementation uses open-loop trajectories with task decomposition; integrating reactive control could enhance robustness in dynamic environments. While we use affordance and avoidance maps for trajectory planning, more sophisticated collision avoidance mechanisms could ensure safer robot-environment interactions. Additional challenges include efficient memory organization for lifelong learning and handling diverse sensory inputs like images and audio.

In conclusion, EnvBridge demonstrates the value of utilizing information from various environments to improve adaptability to new settings and flexible planning capabilities. Addressing these limitations while building upon the current strengths of cross-environment knowledge transfer will be crucial for advancing the practical use of AI agents in diverse real-world scenarios.

## ETHICS STATEMENT

We have read the ICLR Code of Ethics and ensured this paper follows it. Our work does not involve the release of any new datasets; all benchmark datasets used in this research, specifically RLBench, MetaWorld, and CALVIN, are publicly available. We believe our work will have a positive societal

impact by enhancing the adaptability and robustness of robotic manipulation agents, which can lead to more efficient and safer autonomous systems in various real-world applications.

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

# A APPENDIX

## A.1 PARAMETERS FOR EVALUATION

Table 3: Parameters

| Parameters for EnvBridge | |
| --- | --- |
| Text Embedding Model | sentence-transformers/all-MiniLM-L6-v2 |
| LLM model for RLBench and CALVIN evaluation | gpt-4o-mini-2024-07-18 |
| LLM model for MetaWorld evaluation | gpt-4o-2024-05-13 |
| LLM temperature | 0 |
| LLM max token | 512 |

## A.2 PROMPTS IN RLBENCH

The following are the prompts used in RLBench. Also, {} indicates variables where examples and other content will be inserted.

### A.2.1 PROMPT FOR CODE GENERATION

The prompt is exactly the same as VoxPoser.

**Instruction**
You are a helpful assistant that pays attention to the user's instructions and writes good python code for operating a robot arm in a tabletop environment.

**User**
I would like you to help me write Python code to control a robot arm operating in a tabletop environment. Please complete the code every time when I give you new query. Pay attention to appeared patterns in the given context code. Be thorough and thoughtful in your code. Do not include any import statement. Do not repeat my question. Do not provide any text explanation (comment in code is okay). I will first give you the context of the code below:
{*The same examples as VoxPoser*}
Note that x is back to front, y is left to right, and z is bottom to up.

**Assistant**
Got it. I will complete what you give me next.

**User**
# {*Query*}

### A.2.2 Prompt for Knowledge Transfer

---

**Instruction**

You are a helpful assistant that pays attention to the user's instructions and writes good python code for operating a robot arm in a tabletop environment.

**User**

I would like you to help me write Python code to control a robot arm operating in a tabletop environment. Please complete the Python code to execute the query in the target environment. Pay attention to appeared patterns, coordinates, and scale in the given context code. Be thorough and thoughtful in your code. Do not include any import statement. Do not repeat my question. Do not provide any text explanation. I will first give you the context of the code in the target environment below:
{*The same examples as VoxPoser*}
Note that coordinates and scale should be modified to match the target environment. Please do not generate empty code.

**Assistant**

Got it. I will complete what you give me next.

**User**

# Python code in different environment
{*Example retrieved from memory*}
# Python code in target environment

---

### A.2.3 Prompt for Re-Planning

---

**Instruction**

You are a helpful assistant that pays attention to the user's instructions and writes good python code for operating a robot arm in a tabletop environment.

**User**

I would like you to help me write Python code to control a robot arm operating in a tabletop environment. Please complete the code every time when I give you new query. Pay attention to appeared patterns in the given context code. Be thorough and thoughtful in your code. Do not include any import statement. Do not repeat my question. Do not provide any text explanation (comment in code is okay). I will first give you the context of the code below:
{*The same examples as VoxPoser*}
Note that x is back to front, y is left to right, and z is bottom to up. Please refer to the Python code for similar query.

**Assistant**

Got it. I will complete what you give me next.

**User**

# Python code for similar query
{*Example retrieved and transferred from memory*}
# Python code for target query

---

757

A.2.4 PROMPT FOR SELF-REFLECTION

**Instruction**
You are a helpful assistant that pays attention to the user's instructions and writes good python code for operating a robot arm in a tabletop environment.

**User**
I would like you to help me consider new plan to solve the task in a tabletop environment using a robot arm. Please complete the new plan every time when I give you new Python code. Do not summarize the Python code, but rather think about the strategy and path you took to attempt to complete the task. Devise a concise, new plan of action that accounts for the wrong code with reference to specific actions that you should have taken. I will first give you the examples of the new plan:

objects = ['grape', 'lemon', 'drill', 'router', 'bread', 'tray']
# Query: put the sweeter fruit in the tray that contains the bread.
composer("grasp the grape")
composer("back to default pose")
composer("move to the top of the tray that contains the bread")
composer("open gripper")
# done STATUS: FAIL
New plan: Based on the image, I was able to grasp the grape, but I couldn't place it in the correct position. Therefore, I will create a detailed plan by interpreting the intent of the query regarding where to place it.

objects = ['drawer', 'umbrella']
# Query: close the drawer.
composer("push close the drawer handle by 25cm")
# done STATUS: FAIL
New plan: Based on the image, I couldn't reach the drawer handle. Therefore, the plan should take into account the location of the target and the position of other objects.

Please refer to the given image as the current observation information.

**Assistant**
Got it. I will complete what you give me next.

**User**
{*Generated Python code in the previous trial*}
STATUS: FAIL
New plan:

A.3 MEMORY EXAMPLES FROM RLBENCH

```
1   {
2       "environment": "rlbench",
3       "type": "planner",
4       "query": "pick up the rubbish and leave it in the trash
         ↪  can.",
5       "code": "objects = ['bin', 'rubbish', 'tomato1', 'tomato2']
6       # Query: pick up the rubbish and leave it in the trash can.
7       composer(\"grasp the rubbish\")
8       composer(\"back to default pose\")
9       composer(\"move to the top of the bin\")
10      composer(\"open gripper\")
```

```
11      # done"
12    }

1    {
2        "environment": "rlbench",
3        "type": "composer",
4        "query": "grasp the rubbish.",
5        "code": "# Query: grasp the rubbish.
6    movable = parse_query_obj('gripper')
7    affordance_map = get_affordance_map('a point at the center of
     ↪  the rubbish')
8    gripper_map = get_gripper_map('open everywhere except 1cm
     ↪  around the rubbish')
9    execute(movable, affordance_map=affordance_map,
     ↪  gripper_map=gripper_map)"
10   }
```

## A.4 BENCHMARK DETAILS

### A.4.1 RLBENCH

Table 4: Task-specific success rate(%) on RLBench. Best results are in bold.

| Tasks | Baseline | Retry | Self-Reflection | EnvBridge |
|---|---|---|---|---|
| BasketballInHoop | 15 | 15 | 20 | **25** |
| BeatTheBuzz | 60 | 70 | **90** | 65 |
| CloseDrawer | 60 | 65 | **85** | 75 |
| LampOff | 55 | 75 | **80** | 65 |
| OpenWineBottle | 15 | 40 | 65 | **95** |
| PushButton | 15 | 25 | 45 | **80** |
| PutRubbishInBin | 60 | 80 | **100** | 90 |
| TakeFrameOffHanger | 5 | 15 | **30** | 15 |
| TakeLidOffSaucepan | 0 | 0 | 15 | **85** |
| TakeUmbrellaOutOfUmbrellaStand | 80 | 85 | **95** | **95** |
| Average | 36.5 | 47.0 | 62.5 | **69.0** |

### A.4.2 MEMORY COMPARISON ON RLBENCH

Through comparison of various source environment memories, using CALVIN s memory yields the best performance. It shows superior results compared to using RLBench memory, which is the same environment as the evaluation setting, with particularly significant improvements in the PushButton and TakeLidOffSaucePan tasks.

This finding suggests that for challenging tasks, leveraging knowledge from different environments can potentially achieve better results than using knowledge from the same environment, even if the latter contains more examples.

### A.4.3 METAWORLD

We compare different memory settings in section 4.2. We focuse on RLBench as the source environment for transferred knowledge results because MetaWorld's standardized task set and simulated table robotic manipulation scenario are similar to the settings in RLBench. Here, we add CALVIN memory for transferred knowledge into comparison.

Table 5: Task-specific success rate(%) on RLBench with different memory. Best results are in bold.

| Tasks | Retry | EnvBridge (MetaWorld) | EnvBridge (CALVIN) | EnvBridge (RLBench) |
|---|---|---|---|---|
| BasketballInHoop | 15 | 15 | 25 | **35** |
| BeatTheBuzz | 70 | **75** | 65 | 60 |
| CloseDrawer | 65 | 50 | **75** | 70 |
| LampOff | 75 | 80 | 65 | **85** |
| OpenWineBottle | 40 | 65 | **95** | **95** |
| PushButton | 25 | 40 | **80** | 55 |
| PutRubbishInBin | 80 | **95** | 90 | **95** |
| TakeFrameOffHanger | 15 | **20** | 15 | 10 |
| TakeLidOffSaucepan | 0 | 40 | **85** | 55 |
| TakeUmbrellaOutOfUmbrellaStand | 85 | **95** | **95** | **95** |
| Average | 47.0 | 57.5 | **69.0** | 65.5 |

The results show that knowledge transfer from CALVIN (31% average success rate) performs slightly worse than transfer from RLBench (37% average success rate) when applied to MetaWorld tasks. This could be explained by the factor that RLBench's environment and task structure might be more similar to MetaWorld compared to CALVIN, making knowledge transfer more effective.

Table 6: Task-specific success rate(%) on MetaWorld

| Tasks | Baseline | In-domain | Transferred (RLBench) | Transfoerred(CALVIN) | Unified |
|---|---|---|---|---|---|
| Hammer | 25 | 35 | 25 | 25 | **50** |
| Window-close | 20 | **55** | 35 | 25 | 50 |
| Assembly | 25 | **50** | 35 | 35 | **50** |
| Button-Press | 20 | 35 | 40 | 35 | **55** |
| Drawer-Open | 35 | 65 | 50 | 55 | **85** |
| Average | 25 | 48 | 37 | 31 | **56** |

## A.5 QUALITATIVE ANALYSIS ON RLBENCH

We provide specific success and failure examples as part of our qualitative evaluation. In the success case, the Planner generates detailed subtasks. In contrast, in the failure case, it produces almost identical subtasks. We believe this issue arises because the memory lacks the appropriate code to solve the tasks, preventing the presentation of suitable examples.

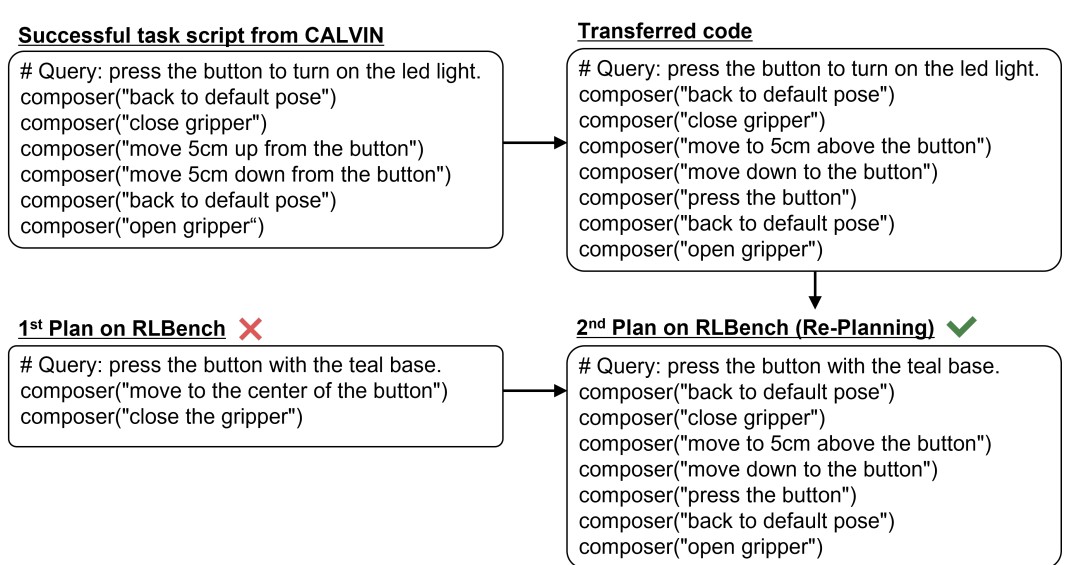

Figure 7: Success Case

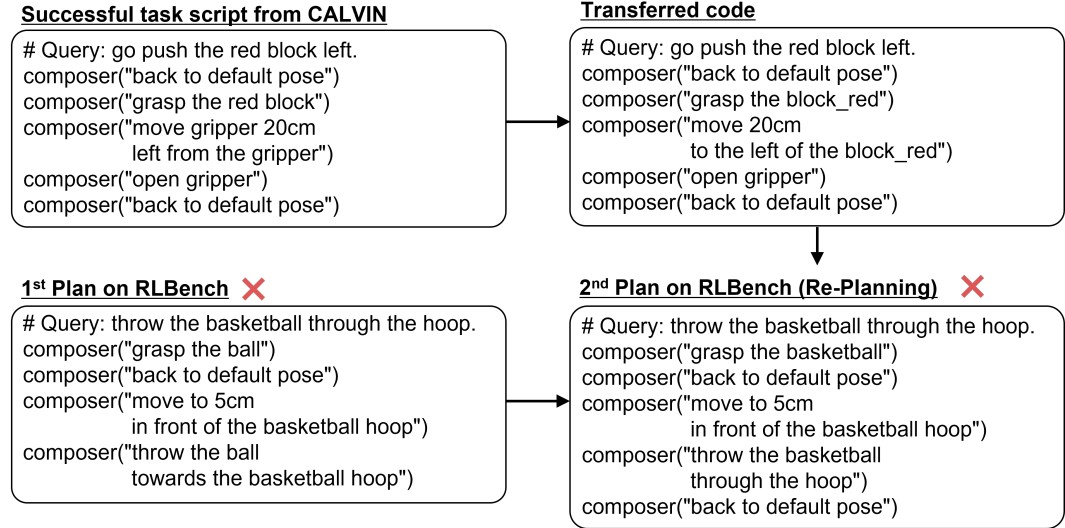

Figure 8: Failure Case

