# OpenReview forum: "EnvBridge: Bridging Diverse Environments with Cross-Environment Knowledge Transfer for Embodied AI"
_ICLR.cc/2025/Conference — Submitted to ICLR 2025_

### Official Review · Reviewer_2Mrc · 2024-10-26

**Soundness:** 2
**Presentation:** 3
**Contribution:** 2
**Rating:** 3
**Confidence:** 3

**Summary:**

The paper introduces ENVBRIDGE, a method that leverages Large Language Models (LLMs) for cross-environment knowledge transfer in robotic manipulation. The core idea is to use the control code from source environments as prompts to generate suitable control actions in new target environments. The proposed approach is tested across three common robotic manipulation benchmarks: RLBench, MetaWorld, and CALVIN.

**Strengths:**

1. The use of LLMs for cross-environment task transfer is a creative and intriguing approach. I especially like the method of using previously successful control code as in-context prompts for generating control code in new tasks.
2. The experiments are conducted across three well-established manipulation benchmarks.

**Weaknesses:**

1. My primary concern is the practical applicability of the method. Why should we rely on LLMs to generate control code for simulated environments, and how feasible is it to extend this approach to real-world robotic systems? As far as I understand, the method depends on ground-truth states and simulation-specific API calls, which may not easily translate to real-world scenarios where such information is not readily available.
2. Certain aspects of the experimental setup are not clearly explained. I have raised specific questions about this below.

**Questions:**

1. Could the authors clarify whether the method uses ground-truth states as input or camera inputs for perception?
2. The experimental protocol in Section 4.1 is somewhat ambiguous. In lines 340–341, the paper mentions “We sampled 10 tasks from it and conduct evaluations covering various tasks, instructions, and objects.” Does this imply different variations were introduced within these 10 tasks? More explanation would help.
3. In Section 4.2, why didn’t the authors use CALVIN as a source task, similar to the approach in Section 4.1?
4. In Section 4.3, what were the source tasks used for this experiment? Were they drawn from RLBench and MetaWorld?
5. The paper only compares ENVBRIDGE with VoxPoser, yet there are other related works that also utilize LLMs for generating robot control code. How does ENVBRIDGE compare to methods like Code as Policies?
6. Some minor formatting issues exist in the paper, such as mismatched quotation marks in lines 192 and 193.

---

> ### Author Response · Authors · 2024-11-18
> **Reply to Reviewer 2Mrc(1/2)**
>
> Thank you for your valuable feedback and for recognizing the creative potential of our approach. We particularly appreciate your positive feedback on using previously successful control code as in-context prompts for new tasks. This is indeed a key innovation of our work.
>
> We address your concerns and questions below.
>
> > Weakness 1: Practical applicability to real-world robotic systems
>
> Thank you very much for your insightful comment regarding the real-world applicability of our approach.
>
>
>
> Existing methods such as CaP and VoxPoser have successfully demonstrated the feasibility of LLM-based robot code generation for real robots. Building upon VoxPoser's framework, our approach introduces Cross-Environment Knowledge Transfer techniques to enhance adaptability across various environments, thereby extending its potential for real-world applications.
>
>
>
> While our current implementation relies on ground-truth observation data in simulation environments, we can extend our system to real-world applications by following VoxPoser's proven approach. Specifically, VoxPoser has demonstrated that Vision Language Models (VLMs) like OWL-ViT can effectively replace simulation-based observations with real visual information processing. Through this integration, our system can transition to real-world deployment without depending on simulation-specific APIs.
>
> > Question 1: Perception Choice
>
> Thank you for your question regarding our system's input modalities.
>
> In our current evaluation, we utilize ground-truth states as input. However, for real-world applications, our system is designed to work with camera inputs processed through Vision Language Models (VLMs), ensuring practical deployability beyond simulation environments.
>
> > Question 2: The experimental protocol in Section 4.1
>
> Thank you for your question about our experimental protocol.
>
>
>
> We selected 10 RLBench tasks (detailed in section A.4.1) and evaluated each task across 20 episodes with randomized instructions, objects, and placements. The success rate was calculated by averaging these 20 episodes per task, following the same evaluation protocol as VoxPoser.

---

> ### Author Response · Authors · 2024-11-18
> **Reply to Reviewer 2Mrc(2/2)**
>
> > Question 3: Metaworld results with CALVIN knowledge
>
> We focused on RLBench as the source environment because MetaWorld’s standardized task set and simulated table robotic manipulation scenario are similar to the settings in RLBench. However, we sincerely agree with your suggestion to include CALVIN results, which is crucial for more comprehensive comparison. Here we have the updated results.
>
>
> | Tasks |  Baseline | In-domain | Transferred (RLBench) | Transferred (CALVIN) | Unified |
> | -------------- |----------|------------|----------------------|---------------------|---------|
> | Hammer       | 25       | 35        | 25                   | 25                 | **50**      |
> | Window-close | 20       | **55**        | 35                   | 25                 | 50      |
> | Assembly     | 25       | **50**        | 35                   | 35                 | **50**      |
> | Button-Press | 20       | 35        | 40                   | 35                 | **55**      |
> | Drawer-Open  | 35       | 65        | 50                   | 55                 | **85**      |
> | Average      | 25       | 48        | 37                   | 31                 | **56**      |
>
>
> The new results show that knowledge transfer from CALVIN (31% average success rate) performs slightly worse than transfer from RLBench (37% average success rate) when applied to MetaWorld tasks. This could be explained by the factor that RLBench's environment and task structure might be more similar to MetaWorld compared to CALVIN, making knowledge transfer more effective.
>
> > Question 4: Section 4.3 source tasks
>
> Thank you for pointing out the evaluation settings. We use RLBench as source tasks for CALVIN evaluation, and we will add this information to our rebuttal revision.
>
>  > Question 5: Baseline choices
>
> EnvBridge utilizes VoxPoser as its code generation module, and we compared these systems to evaluate the improvements achieved through our proposed Cross-Environment Knowledge Transfer. We chose VoxPoser over other methods such as Code as Policies (CaP) primarily because VoxPoser's hierarchical structure and higher-level planning modules are more suitable for EnvBridge.
>
>
>
> Additionally, the previous study [1] has reported the following evaluation results of CaP and VoxPoser on RLBench tasks:
>
> | Task | CaP | VoxPoser | (EnvBridge) |
> | ---- | ---- | ---- | ---- |
> | Take_umbrella | 4.0 | 33.3 | (95.0) |
> | Open_wine | 0 | 8.0 | (95.0) |
>
> While direct comparisons with EnvBridge are not feasible due to differences in experimental conditions, existing comparative studies between VoxPoser and CaP have demonstrated VoxPoser's superior performance.
>
> [1] J. Duan, W. Yuan, W. Pumacay, Y. R. Wang, K. Ehsani, D. Fox, and R. Krishna, “Manipulate-anything: Automating real-world robots using vision-language models,” arXiv preprint arXiv:2406.18915, 2024.
>
> > Question 6: Minor formatting issues
>
> Thank you for pointing out the formatting issues, we will update accordingly in the rebuttal revision.

---

> > ### Comment · Reviewer_2Mrc · 2024-11-22
> >
> > Thank you for providing a detailed rebuttal. While I appreciate the authors’ efforts to address the concerns raised, I remain unconvinced by their response regarding the work’s applicability to real-world scenarios.
> >
> > As noted in the rebuttal, VoxPoser already includes a pipeline for implementing vision-based policies, which makes it more practical for real-world extensions. However, this work builds heavily on VoxPoser, using it as a central baseline, yet does not present any vision-based experiments. I believe that incorporating such experiments would significantly enhance the impact of this work. While real-world experiments would be ideal, I understand that for ML venues like ICLR, they are not strictly necessary. However, the absence of vision-based experiments represents a notable weakness.
> >
> > As a result, I will maintain my current evaluation.

---

> > > ### Author Response · Authors · 2024-11-26
> > > **Reply to Official Comment by Reviewer 2Mrc**
> > >
> > > Thank you for your thoughtful feedback and we would like to further clarify our contribution.
> > >
> > > High-level planning and low level execution are two separate critical aspects of robot manipulation. Our work's primary contribution lies in improving high-level planning through Cross-Environment Knowledge Transfer, which operates independently of the perception system. The effectiveness of our method in improving transfer learning capabilities would remain valid regardless of whether using vision-based or ground-truth object information, as the core mechanism operates at the planning level. We provide vivid examples in Figure 7 of how our proposed method can improve embodied agent's planning ability, thus improving task successful rates.
> > >
> > > To ensure fair evaluation, we use the same ground-truth object information as provided in VoxPoser's public implementation, as their vision-based implementation is not publicly available. This approach allows us to evaluate the effectiveness of our method in a controlled manner, focusing specifically on the language-based transfer mechanisms.
> > >
> > > While we acknowledge that real-robot validation is important in robotics conferences, we believe our theoretical contribution regarding the transferability of LLM Agents is well-suited for machine learning venues like ICLR, where the emphasis is on advancing fundamental learning mechanisms.  EnvBridge is robust across various environments with Cross-Environment Knowledge Transfer, avoiding specific training or adjustments.
> > >
> > > We still remain committed to enhancing the clarity and comprehensiveness of our work. We would appreciate it if you could reconsider the ratings. Please let us know if you have lingering questions.

---

### Official Review · Reviewer_89oP · 2024-11-02

**Soundness:** 1
**Presentation:** 1
**Contribution:** 1
**Rating:** 3
**Confidence:** 3

**Summary:**

This paper proposes EnvBridge, an LLM-based method that can leverage diverse knowledge to solve complex tasks. It contains several components to endow the agents with good generalization ability.

**Strengths:**

The proposed methods are evaluated on 3 benchmarks.

**Weaknesses:**

- The Introduction section includes many discussions not related to LLMs for robotic manipulation, which appears redundant.

- The writing of the article is not very clear; concepts such as cross-embodiment and cross-environment need to be more distinctly defined. This significantly detracts from the overall logic of the paper.

- The quality of the figures in the paper is poor.

- I do not agree with the author's assertion that current LLM-based methods have significant generalization issues. Many LLM-based approaches have demonstrated strong generalization abilities in real-world applications. I also do not believe that the method proposed by the author addresses the existing issues effectively.

- I believe that this paper is not yet ready for submission to ICLR.

**Questions:**

Seen above.

---

> ### Author Response · Authors · 2024-11-15
> **Reply to Reviewer 89oP**
>
> Thank you very much for taking the time to review our paper and providing your feedback.
> We would like to address each of your comments in detail below.
>
> > Weakness 1: Introduction Section Redundancy
>
>    Thank you for your opinions. We understand that extraneous information can lead to confusion.
>    Our introduction flows from the general landscape of LLMs and recent developments, followed by a transition to robotic manipulation as a use case that is gaining traction, and then to limitations of current works in the area before proposing our solution to tackle these limitations.
>    We would appreciate it if you could let us know which points you found redundant.
>
> > Weakness 2: Clarity of the Article
>
>    We appreciate this feedback.
>    We will consider revising the sentences to provide detailed and clear definitions.
>
> > Weakness 3: Quality of the Figures
>
>    We acknowledge that clear figures can greatly enhance comprehension.
>    We would be grateful if you could let us know which aspects of the figures you found concerning.
>
> > Weakness 4: Disagreement on Generalization Issues
>
>    We sincerely thank the reviewer for raising this important point about generalization capabilities. We would like to respectfully offer some clarification about our specific findings:
>
>    4.1. Our paper identifies two specific limitations in current LLM-based approaches:
>    - Policy-based methods require environment-specific training, limiting cross-environment transfer
>    - Code generation methods struggle with fixed prompts in unseen environments
>
>    4.2. Our experiments systematically examined these limitations:
>    - The baseline tests show VoxPoser achieves 36.5% success on RLBench and 25% on MetaWorld tasks
>    - These results suggest there may still be room for improvement in handling unseen environments
>
>    4.3. Our proposed approach shows promising progress in addressing these challenges:
>    - Improving success rate to 69.0% on RLBench
>    - Achieving 56% success rate on MetaWorld
>
>    We would be very interested to learn about the specific LLM-based approaches the reviewer has in mind that demonstrate strong generalization abilities in robotic manipulation. This would be valuable for our research and help us provide more comprehensive comparisons. We are always eager to learn from and build upon successful approaches in the field.
>
> Once again, thank you for your valuable feedback and time.
> To conduct better research, we would be grateful if you could provide more detailed advice, if possible.

---

> > ### Comment · Reviewer_89oP · 2024-11-23
> > **reply to author**
> >
> > Thanks for your reply. For Weakness 1, The first paragraph in the introduction could be removed; For Weakness 3, the figure 3 & 4 occupy half of the page. Also, there are a lot of follow-up works after VoxPoser, such as Recap, VILA, Copa etc.  You should let the reader know what's the difference between your work and the others. Why your method has the advantages compared to their implementation. Meanwhile, as Reviewer 2Mrc, I totally agree that it is important to incorporate vision-based experiments. The real-world experiments are also recommended to enhance the paper. I will keep my rate.

---

> > > ### Author Response · Authors · 2024-11-26
> > > **Reply to Official Comment by Reviewer 89oP**
> > >
> > > Thank you for your response. We appreciate your suggestions and would like to address each point carefully.
> > >
> > > > The first paragraph in the introduction
> > >
> > > We understand your suggestion regarding the first paragraph. However, we believe this background on LLM reasoning capabilities and Chain-of-Thought provides important context for our work, as our method fundamentally builds upon these concepts to enhance embodied agents' planning abilities through Cross-Environment Knowledge Transfer.
> > >
> > > > Figure 3 and Figure 4
> > >
> > > Thank you for your advice. We will further refine the layout of our paper.
> > >
> > > > More related work
> > >
> > > Thank you for bringing more related work to attention. We have carefully considered how our work relates to these approaches:
> > >
> > > - ReKep presents advances in reactive robot manipulation through its retry mechanism. While ReKep excels at handling individual task failures, EnvBridge takes a different approach by focusing on cross-environment knowledge transfer and leveraging past successful experiences stored in memory for continuous learning. These approaches could be complementary, combining ReKep's reactive capabilities with EnvBridge's broader knowledge transfer framework.
> > >
> > > - Copa focuses primarily on low-level control aspects of robot manipulation, while our work addresses high-level planning capabilities. While both contribute valuable advances, they address different aspects of the robotics pipeline and could potentially be complementary.
> > >
> > > - Regarding ViLA, we acknowledge its contributions to the field. However, due to its focus on real-world environments and lack of available open-source code, direct experimental comparison was not feasible within our evaluation framework.
> > > We will expand our related work discussion to better position our approach within this broader context.
> > >
> > > > Vision-based scenarios.
> > >
> > > We appreciate this important point about vision-based evaluation. High-level planning and low-level execution are two separate critical aspects of robot manipulation. Our work's primary contribution lies in improving high-level planning through Cross-Environment Knowledge Transfer, which operates independently of the perception system. The effectiveness of our method in improving transfer learning capabilities would remain valid regardless of whether using vision-based or ground-truth object information, as the core mechanism operates at the planning level. We provide vivid examples in Figure 7 of how our proposed method can improve embodied agent's planning ability, thus improving task successful rates.
> > >
> > > We still remain committed to enhancing the clarity and comprehensiveness of our work. We would appreciate if you could reconsider the ratings. Please let us know if you have lingering questions.

---

### Official Review · Reviewer_d9n1 · 2024-11-03

**Soundness:** 4
**Presentation:** 3
**Contribution:** 2
**Rating:** 6
**Confidence:** 3

**Summary:**

This work presents EnvBridge, a novel approach for computing robot trajectories via LLM-prompted code generation. The authors argue that transferring code across different environments, similar to human reasoning, can improve the success rate of task completion. EnvBridge extends existing methods like VoxPoser by adding a knowledge transfer mechanism to bridge across environments.

This process leverages a hierarchical code generation approach, dividing the task into subtasks (Planner) and generating 'language model programs' (LMPs) for each subtask (Composer). These LMPs produce value maps that guide the generation of an open-loop trajectory. I believe this trajectory is for the end-effector pose.

The system operates in several stages. First, an LLM generates code that outputs a trajectory, based on a language query describing the task. This is the same as VoxPoser. If unsuccessful, EnvBridge searches a repository for similar tasks where code generation failed at the same stage (Planner or Composer). Successful code from a similar task is then transferred to the target environment and used as an example in a revised prompt for the LLM, known as Knowledge Transfer and Re-planning.

**Strengths:**

The primary strength lies in its novel approach to knowledge transfer for robot code generation via appropriate prompt engineering. The idea is well-motivated by observations of human behavior and offers a promising direction for improving LLM performance in robotics. The paper is generally well-written and presents the approach with clarity. Furthermore, the authors conduct extensive experiments across multiple tasks and environments in simulation, demonstrating the effectiveness of EnvBridge.

**Weaknesses:**

1. EnvBridge generates open-loop trajectories, lacking the reactivity necessary for dynamic environments.
2. To my understanding, the system focuses solely on end-effector positions, neglecting possible collisions of the robot arm with the environment.
3. The user must explicitly provide information about objects to avoid in the scene, which could be automated through prompt engineering. For example, in the example memory shown in section A.3, the task is to "pick up the rubbish and leave it in the trash can.", and there are a bunch of objects given. The Planner level should create subtasks to avoid other objects (e.g. 'tomato1') while picking up the object named 'rubbish'.
4. The evaluation could be strengthened by comparing EnvBridge to other baselines like OpenVLA, which also generates end-effector poses from similar inputs.
5. The authors should provide further clarification regarding the performance discrepancies observed in Figure 2, where code from the same environment (RLBench) performs worse than transferred code from a different environment (CALVIN) - even though RLBench memory has more code examples than those in CALVIN.
6. It would be good to have real-robot experiments, to improve the case for accepting any robotics paper.

Finally, to best understand the work, you need to see the prompts given in the appendix, and the type of prompt engineering used for each stage of EnvBridge. While this is not explicitly a weakness, these are important details worthy of being in the main paper.

### Minor Points

A possible typo on line 57: "transfer ability" should likely be "transferability"??
Throughout the paper (e.g., line 694), the authors should explicitly clarify that curly braces in the prompts `{}` denote variables to be filled in.

**Questions:**

See above

---

> ### Author Response · Authors · 2024-11-19
> **Reply to Reviewer d9n1 (1/2)**
>
> Thank you for your thorough and constructive feedback! We are particularly encouraged by your recognition of our novel approach Cross-Environment Knowledge Transfer for robot code generation and its clear motivation from human behavior. We also appreciate your acknowledgment of our extensive experimental validation across multiple tasks and environments, which we indeed invested significant effort in to demonstrate the effectiveness of EnvBridge.
>
> We address your concerns and questions below.
>
> > Weakness 1: Open-loop Trajectories
>
> We appreciate your thoughtful observation about open-loop trajectories in dynamic environments. We would like to respectfully offer some additional context about our approach that may help clarify this aspect of our work.
> - First, while EnvBridge does generate open-loop trajectories, it's important to note that these trajectories are generated separately for each decomposed subtask, not for the entire task at once. Our Planner and Composer break down complex tasks into simpler subtasks, each requiring only short, straightforward trajectories. This decomposition significantly reduces the risk of execution failures.
> - Second, we sincerely agree that reactive control is crucial for robust operation in dynamic environments. However, EnvBridge's primary contribution is improving the agent's planning capabilities through cross-environment knowledge transfer, which is orthogonal to reactive control. We believe our approach is complementary to reactive control methods - the enhanced planning capabilities from cross-environment knowledge transfer could be combined with reactive control techniques to create more robust systems. We have added this discussion in our rebuttal revision to clarify the scope of our contribution and potential future directions.
>
>
> > Weakness 2: Addressing collision with environment
>
>   Thank you for pointing this out. The paper is inspired from Voxposer, where they use the concept of affordance and avoidance maps, for entity of interest and entities to be repulsed after which the trajectory is planned. This allows the system to consider potential collisions with the environment and plan trajectories that minimize such interactions. By leveraging these techniques, we aim to ensure safe and efficient robot behavior.
>
>
> Regarding the concern about potential collisions involving other parts of the robot (e.g., arms or base) with the environment, we acknowledge this as a valid limitation. Addressing this challenge is an important direction for future work. We plan to explore and implement improved methods that account for the entire robot’s structure, incorporating advanced collision-avoidance techniques to enhance the system's overall safety and effectiveness in complex environments. Thank you for bringing this to our attention.
>
>
> > Weakness 3: Ensuring Avoidance of Non-Target Entities
>
> Thank you for your comment. Currently we have considered the avoidance maps at composer level. For example, if the planner generates the following query from the perception received, the composer generates the relevant affordance and avoidance maps:
>
> ```
> # Query: move to the back side of the table while staying at least 5cm from the blue block.
> movable = parse_query_obj('gripper')
> affordance_map = get_affordance_map('a point on the back side of the table')
> avoidance_map = get_avoidance_map('5cm from the blue block')
> execute(movable, affordance_map=affordance_map, avoidance_map=avoidance_map)
> ```
>
> The composer generates maps to ensure the gripper interacts only with intended entities while avoiding unnecessary contact with others, such as the blue block. This method effectively minimizes unintended interactions, ensuring a safer and more efficient trajectory execution.
>
>
> > Weakness 4: Baseline comparison
>
> Thank you for the suggestion. Comparing EnvBridge with other baselines is an interesting direction. We have compared our work to RVT—a multi-view transformer for 3D manipulation that utilizes RLBench training dataset with 100 expert demonstrations per task used to predict robot end-effector pose. EnvBridge achieves an average success rate of 69% on the RLBench benchmark compared to RVT’s 62.9%, highlighting the effectiveness of our approach. While we acknowledge that the tasks used in RVT’s evaluation are not identical to ours, this comparison highlights the efficiency of our approach. What sets EnvBridge apart is its domain-agnostic nature. Unlike RVT, which requires domain-specific training, EnvBridge is designed to generalize across diverse environments and tasks without extensive retraining or adaptation. This versatility makes EnvBridge both effective and adaptable, offering broader usability compared to baselines that are tied to specific domains. This distinction underscores the potential of our approach to address a wider range of applications in dynamic and varied settings.

---

> > ### Author Response · Authors · 2024-11-19
> > **Reply to Reviewer d9n1 (2/2)**
> >
> > > Weakness 5: Performance Discrepancies
> >
> > Thank you for your insightful comment.
> > Since the comment is about memory comparison, we respond based on our understanding that you are referring to Table 2 rather than Figure 2. However, if our understanding is not correct, we would appreciate it if you could let us know.
> > We found that code from the CALVIN environment demonstrated better performance than code from RLBench, primarily due to CALVIN's diverse set of manipulation patterns that are necessary for specific task solving. To address your comment directly, we conducted a detailed analysis of the success rates for each task:
> > | Task | Retry | EnvBridge(RLBench) | EnvBridge(CALVIN) |
> > | ---- | ---- | ---- | ---- |
> > | BasketballInHoop | 15 | 35 | 25 |
> > | BeatTheBuzz | 70 | 60 | 65 |
> > | CloseDrawer | 65 | 70 | 75 |
> > | LampOff | 75 | 85 | 65 |
> > | OpenWineBottle | 40 | 95 | 95 |
> > | PushButton | 25 | 55 | 80 |
> > | PutRubbishInBin | 80 | 95 | 90 |
> > | TakeFrameOffHanger | 15 | 10 | 15 |
> > | TakeLidOffSaucepan | 0 | 55 | 85 |
> > | TakeUmbrellaOutOfUmbrellaStand | 85 | 95 | 95 |
> >
> > Our analysis reveals a notable pattern: several tasks where the baseline (VoxPoser) showed poor performance (specifically, PushButton and TakeLidOffSaucepan) demonstrated remarkable improvements when utilizing CALVIN memory.
> > This finding suggests that for challenging tasks, leveraging knowledge from different environments can yield better results than using knowledge from the same environment, even when the latter contains more examples.
> >  We will clarify this important insight in the revised manuscript.
> >
> >
> > > Weakness 6: Real-world robot testing
> >
> > Thank you for your comment. We will consider evaluations on real robots too.
> >
> > > Minor Points:  Formatting and typographical errors
> >
> > Thank you for your comments, we will revise our paper based on these points for the camera-ready version.

---

> > > ### Comment · Reviewer_d9n1 · 2024-11-23
> > >
> > > Thank you for your responses and comments.
> > >
> > > >  Weakness 3: Ensuring Avoidance of Non-Target Entities
> > > > Currently we have considered the avoidance maps at composer level. For example, if the planner generates the following query from the perception received, the composer generates the relevant affordance and avoidance maps
> > >
> > > And you gave an example query
> > >
> > > > Query: move to the back side of the table while staying at least 5cm from the blue block.
> > >
> > > If the planner already knows that object 'blue block' exists, can you ensure that it would always plan to avoid objects not mentioned by the user -- without being explicitly told by the user to do so? This is to enable scalability of your method, for instances when you have many different objects in the scene but only 1-2 are referred to in the instructions.

---

> > > > ### Author Response · Authors · 2024-11-25
> > > > **Reply to Official Comment by Reviewer d9n1**
> > > >
> > > > Thank you for your insightful comment. We agree with your observation. In environments such as RLBench, the planner can be provided with a complete list of objects in the scene, enabling it to plan effectively and ensure that non-target entities are consistently avoided. However, in environments with a large number of objects, avoiding all entities may become challenging. In such cases, perception-based information could be utilized to identify and avoid specific objects with collision potential. Addressing this limitation is an area for future work.

---

### Author Response · Authors · 2024-11-22
**Response to Reviews and Summary of Rebuttal Revision**

Dear Reviewers,

We thank you all for your time and effort investing in the review process. We particularly appreciate reviewer d9n1's recognition of our work's soundness (rated as excellent/4) and the positive assessment of our presentation (rated as good/3) by reviewers d9n1 and 2Mrc. We are encouraged that reviewers d9n1 and 2Mrc have recognized the novelty of our approach to cross-environment knowledge transfer for robot code generation through prompt engineering.

We have carefully addressed each reviewer's concerns and questions in our individual responses below their respective reviews. Additionally, we have made several substantial improvements to the paper in our rebuttal revision:

- Fixed some typos and minor formatting issues, including typos in Line 57, 694 [addressing Reviewer d9n1] and formatting issues in Line 192, 193 [addressing Reviewer 2Mrc].

- Incorporated more comprehensive analysis regarding the results reported in Table 2. [addressing Reviewer d9n1 Weakness 5]

- Included new experimental results in Appendix A.4.3 investigating different knowledge sources. [addressing Reviewer 2Mrc Question 3]

- Expanded the discussion section with more discussion on the influence of open-loop trajectories and potential collisions with environment. [addressing Reviewer d9n1 Weakness 1,2]



We believe these substantial improvements address the key concerns raised in the reviews and strengthen our paper considerably. We welcome any additional questions and will provide further clarifications during the discussion period. We would be grateful if you could consider raising your ratings if you find our revisions satisfactory.

Best regards,

[Authors of Paper #4278]

---

### Meta-Review · Area_Chair_hhct · 2024-12-26

**Metareview:**

The authors propose ENVBRIDGE, an approach which uses large language models for cross-environment transfer for robotics. They transfer robotics control code between environments using the LLMs. They use three different simulators to show that the LLMs can be used to accomplish diverse tasks across these simulations. The system generates end-effector trajectories, similar to prior works like VoxPoser or PerAct, which improves generalization.

Strengths:
Using LLMs to transfer control code in this way is a novel idea.
They use three different well-established simulators: Meta World, RLBench, and Calvin.

Weaknesses:
Only these three simulators, and no real world results
No clear learning contribution
The work would be much stronger with vision experiments (noted by 2Mrc and 89oP)
How much prompt engineering is necessary for adaptation? For example,
More baselines (the authors mention RVT in the discussion by the authors) would make results much more convincing

Overall, this work proposes some very interesting ideas, but these should be validated more strongly.

**Additional Comments On Reviewer Discussion:**

There was a discussion as to whether or not real-world experiments are necessary. For an ML venue, these do not seem important; but many reviewers (2Mrc and 89oP) point out that vision-based policy experiments would both be fairly straightforward (already implemented by their baselines) and would add tremendously to the value of the paper.

Authors raised concerns about wider applicability of the method, and about the number of assumptions the authors are making (especially scenes with unknown objects). These could be addressed through future experiments and more comparisons.

---

### Decision · Program_Chairs · 2025-01-22

Reject